# Molecular Mapping of Allergen Exposome among Different Atopic Phenotypes

**DOI:** 10.3390/ijms241310467

**Published:** 2023-06-21

**Authors:** Ruperto González-Pérez, Paloma Poza-Guedes, Fernando Pineda, Tania Galán, Elena Mederos-Luis, Eva Abel-Fernández, María José Martínez, Inmaculada Sánchez-Machín

**Affiliations:** 1Allergy Department, Hospital Universitario de Canarias, 38320 Tenerife, Spain; pozagdes@hotmail.com (P.P.-G.); elenamederosluis@gmail.com (E.M.-L.); zerupean67@gmail.com (I.S.-M.); 2Severe Asthma Unit, Hospital Universitario de Canarias, 38320 Tenerife, Spain; 3Inmunotek SL Laboratories, 28805 Madrid, Spain; fpineda@inmunotek.com (F.P.); tgalan@inmunotek.com (T.G.); efernandez@inmunotek.com (E.A.-F.); mjmartinez@inmunotek.com (M.J.M.); 4Allergen Immunotherapy Unit, Hospital Universitario de Canarias, 38320 Tenerife, Spain

**Keywords:** airborne allergens, exposome, climate change, atopy, allergic rhinitis, asthma, atopic dermatitis

## Abstract

Climate change and exposure to environmental pollutants play a key role in the onset and aggravation of allergic diseases. As different climate-dependent patterns of molecular immunoglobulin E (IgE) reactivity have been regionally described, we sought to investigate the evolving allergen exposome in distinctive allergic phenotypes and subtropical weather conditions through a Precision Allergy Molecular Diagnosis (PAMD@) model. Concurrent sensitization to several house dust mites (HDM) and storage mite molecules were broadly dominant in the investigated cohort, followed by the major cat allergen Fel d 1, and regardless of the basal allergic disease. Although a complex repertoire of allergens was recognized, a steadily increasing number of IgE binding molecules was associated with the complexity of the underlying atopic disease. Besides the highly prevalent IgE responses to major HDM allergens, Der p 21, Der p 5, and Der p 7 also showed up as serodominant molecules, especially in subjects bothered by asthma and atopic dermatitis. The accurate characterization of the external exposome at the molecular level and their putative role as clinically relevant allergens is essential to elucidate the phenotypic diversity of atopic disease in terms of personalized diagnosis and therapy.

## 1. Introduction

The definition of exposome was coined in 2005 to describe the totality of environmental exposures—from a variety of external and internal sources—that an individual experiences from conception throughout life [1,2]. The more recent concept of the meta-exposome refers to the bidirectional relationship of environmental exposure on human health and the influence of human health on other living systems and genomes [3]. Along with other environmental agents, including chemical, biological, occupational, and physical exposures, aeroallergens are considered part of the specific external exposome [4,5,6]. In fact, during the last few decades a sharp increase in the prevalence of atopic conditions and other immunomediated diseases has been assessed, turning into a relevant health and global economic burden [7,8]. In the era of precision medicine, allergic respiratory and skin diseases can nowadays be grouped according to a combination of clinical, biological, and physiological features into specific pheno-endotypes, paving the road to targeted therapeutics [9,10].

Both epidemiological and environmental data support that the interaction of climate change and air pollutants—including airborne allergens—plays a major role in the pathobiology, evolution, clinical symptoms, and long-term morbidity in human allergic disease [11,12]. Despite aeroallergens being regarded as risk factors for developing allergic disease, the relationship among allergen exposure, sensitization, and clinical symptoms is still a controversial issue [13,14].

The distribution of aeroallergens in a geographical area is highly affected by the local weather conditions, urbanization, and lifestyle, driving diverse types and degrees of specific IgE (sIgE) response dominating different areas of the world [15,16]. In fact, it has been shown that global warming favors warm and humid weather conditions, enhancing a suitable environment for the development of both indoor and outdoor allergens [17,18]. In this regard, as multiple allergens have been described as being capable of sensitizing and inducing clinical manifestations in genetically predisposed subjects, accurate knowledge of the local allergen exposome is mandatory for the prevention, diagnosis, and treatment of allergic diseases [19,20]. Recently, the introduction of Precision Allergy Molecular Diagnosis (PAMD@) has remarkably improved the assay performance—especially in terms of sensitivity and analytic specificity—of a comprehensive assessment of the patient’s sIgE binding to a panel of individual allergens, changing our current understanding of sensitization profiles and potential cross-reactivity [21,22].

Along with these innovations in bioscience and technology, current guidelines propose that allergy diagnostic workup follow a “top-down approach,” primarily from clinical symptoms to specific individual molecules [23]. In addition to former studies evaluating the local prevailing aeroallergens in different regions of Europe [24,25], we aimed to investigate—through a personalized PAMD@ model approach—the aeroallergen exposome and the concurrent sIgE reactivity in three clinical allergic phenotypes under subtropical climate conditions in Tenerife, Spain.

## 2. Results

### 2.1. Demographic Characteristics of Patients

A total of 168 patients were screened, with 150 of them—including 75 females, with a mean age 31.8 year (range 8–67)—finally confirming their eligibility for the study (Figure 1). The patients were divided into three different groups according to their underlying atopic disease (i.e., allergic rhinitis, allergic asthma, or atopic dermatitis). Eighteen patients—8 with rhinitis, 6 with asthma, and 4 with atopic dermatitis—were excluded after a negative skin prick test (SPT) to local aeroallergens (*n* = 10) or current/former allergen immunotherapy (*n* = 18).

All 150 subjects who fulfilled the corresponding clinical practice guidelines on allergic rhinitis (AR), asthma (A), or atopic dermatitis (AD) [26,27,28] showed a positive SPT to one or more of the local aeroallergens and were classified based on the severity of their basal comorbid condition. A Severity Scoring Atopic Dermatitis (SCORAD) index > 40 was considered a marker of severe forms of AD [29]. The majority of patients included in the current investigations were Caucasians (97.3%) and living in urban settlements (77.3%) in the northern sanitary district of the island (Figure 2).

Globally, most of the subjects (114 out of 150 patients, 76.0%) had atopic disease onset during childhood or adolescence. Twenty-seven out of 50 (54%) individuals suffered from severe AR, followed by 44% with severe AD and 42% with severe A. All subjects were on regular daily treatment, comprising both allergen avoidance measures and conventional medical therapy according to their allergic disease stage and severity. All patients (100%) with AR were on treatment with at least daily oral antihistamines. None of the patients were on former or current treatment with allergen immunotherapy or biologics upon inclusion in the current study. Four out 50 (8.0%) patients from the asthma cohort were on oral steroids upon inclusion in the study. Two subjects (4.0%) from the AD cohort were on oral cyclosporine. Overall atopic comorbidities included 23 patients (15.33%) afflicted with a food (seafood, nuts, egg, and/or milk) or drug (7.33%) allergy (beta-lactam antibiotics and/or nonsteroidal anti-inflammatory drugs). Most patients (75.33%) reported a known family history of atopy.

### 2.2. Total IgE and Blood Eosinophils

A quantitative analysis of serum total IgE was performed to evaluate the basal atopic status in the study population. The total IgE showed a median (range) value of 404 (41–19,993) IU/mL. Regarding basal allergic disease, a median (range) value of total IgE of 56 (23.88–1083) IU/mL was found for AR, 674 (56.5–19,993) IU/mL for A, and 898 (81.81–17,420) IU/mL for AD.

Blood eosinophils showed an overall median (range) value of 360 (20–1880) eosinophils/μL, with a slightly higher median value (375 eosinophils/μL) in the AD group with respect to asthmatics (365 eosinophils/μL) and the AR (340 eosinophils/μL) subset (Table 1).

### 2.3. Prevalence, sIgE Reactivity, and Individual Molecular Profile According to Atopic Disease

The sensitization to aeroallergens extracted by SPT and the prevalence of 150 patients who met the inclusion criteria are summarized in Table 2.

One hundred and forty-six patients (97.33%) were independently sIgE positive (≥0.35 kU_A_/L) to one/or more of the individual molecular aeroallergens included in the multiplex array (Figure 3). Despite showing a positive SPT to one or more aeroallergens, 4 subjects (3 afflicted with A and 1 with AD) had an sIgE ≤ 0.35 kU_A_/L to all tested allergens in the multiplex array.

### 2.4. Mites

Mites were identified as the most prevailing source of sensitizing airborne allergens in the present population regardless of the subjects’ atopic disease.

Sensitization to one or more of the 17 investigated mite molecular allergens was found in 144 (96%) subjects. Six patients (1 with AR, 3 with A, and 2 AD) were not sensitized to any of the aforementioned mite molecules. Although all 17 mite allergens were broadly represented throughout the investigated population, significant quantitative differences among molecules and the involved atopic disease were confirmed.

Considering individual molecular allergens exclusively, Der f 2 was most frequently identified with sIgE ≥ 0.35 kU_A_/L in 126 out of 150 subjects (84%) and Der p 2 in 125 patients (83.3%), followed by Der p 23 (82%), Der p 1 (72.6%), Der f 1 (69.3%), and Der p 21 (66.6%). Regarding SM, Lep d 2 was most frequently found in 114 patients (76%), followed by Gly d 2 (67.33%), Blo t 5 (50.6%), Tyr p 2 (45.3%), and Blo t 21 (44%). Eighty-nine out of 150 (59.3%) individuals depicted one or more sIgE molecular responses to Blo t 5, Blo t 21, and/or Blo t 10. Despite no quantitative differences (*p* = 0.411) depicted between the overall mean value of sIgE (kU_A_/L) against Der f 2 (25.67) and Der p 2 (24.12), significant differences (*p* < 0.001) were found among both allergens and Lep d 2 (9.86), Gly d 2 (7.72), and Tyr p 2 (4.18). Significant differences (*p* = 0.014) were also found between the overall mean values (kU_A_/L) of Der p 1 (14.9) and Der f 1 (9.79). A higher frequency of sIgE binding to HDM group 1 (87.03% vs. 66.66%) and 2 (88.88% vs. 79.16%) and Der p 23 (87.3% vs. 84.37%) was found in the younger subjects compared to their elder peers (≥20 years).

Concerning individual mite molecules, Der f 2 (78%), Der p 2 (78%), Der p 23 (76%), Gly d 2 (74%), Lep d 2 (72%), Der p 1 (68%), Blo t 21 (54%), and Der f 1 (52%) were considered serodominant in patients with AR. Blo t 5 was also detected in 46% patients of this group. Minor allergens—Der p 20, Der p 10, Blo t 10, and Der p 11—were identified in <15% of the investigated population.

Seven molecules reached a prevalence >50%—Der f 2, Der p 2, Der p 23, Lep d 2, Der p 1, and Der f 1—in subjects with AR. An increasing number of allergens reaching serodominance was also confirmed for the asthma group—11 individual molecules, including Der p 5 (72%), Der p 7 (58%), and Blo t 5 (56%)—and for those afflicted with AD—12 allergens, including Tyr p 2 (50%). In addition to the large number of sIgE mite responses, increasing quantitative differences from AR to AD were observed among the basal atopic disease and the corresponding mite molecular allergen (Table 3 and Figure 4).

Patients afflicted with AD showed significantly higher serum titers (*p* < 0.05) to eight mite molecules—Der f 2, Der p 2, Der p 23, Der p 1, Der f 1, Der p 21, Der p 5, and Der p 7—than subjects with AR and to four allergens—Der p 23, Der p 1, Der p 21, and Der p 7—compared to the group of asthmatics. Moreover, patients with asthma outlined significantly increased serum titers (*p* < 0.05) of Der p 2, Der f 2, and Der f 1 than individuals with AR. No significant differences were observed among AR, A, or AD in the mean levels of sIgE to Lep d 2, Gly d 2, Blo t 5, Tyr p 2, Blo t 21, Der p 20, Der p 10, Blo t 10, or Der p 11 in the present population. In addition, > 50% of patients with the A and/or AD phenotype showed a more complex aggregation pattern of molecules, including concurrent sensitization to ≥ 10 mite allergens, in contrast to the majority of subjects (64%) with AR displaying a concomitant sIgE response to < 9 molecules (Table 4).

### 2.5. Cat and Dog Epithelia

Globally, 84 out of 150 patients were sensitized to one or more of the 10 investigated epithelial molecular allergens. Considering overall sIgE epithelial response, cat allergens were most frequently identified (43.3%) compared to dog molecules (35.3%) in the studied population.

A total of 65 out of 150 subjects showed different sIgE molecular responses to any of the following cat allergens: Fel d 1 (60.39%), Fel d 7 (19.8%), Fel d 4 (14.85%), and Fel d 2 (3.96%). In addition, a total of 53 out of 150 individuals displayed diverse molecular responses to dog allergens as follows: Can f 5 (46.23%), Can f 1 (22.58%), Can f 4 (20.43%), Can f 6 (15.05%), Can f 2 (8.6%), and Can f 3 (7.52%).

Fel d 1 exhibited as the most prevalent individual epithelial allergen in 19 out of 84 (22.61%) patients, followed by Can f 5 (15.47%). One patient (1.19%) showed a monomolecular sIgE response to Fel d 7. Forty-four subjects out of 150 (29.33%) were simultaneously co-sensitized to two or more cat and dog molecular allergens. In this regard, 84 subjects out of 150 showed a total of 31 heterogenous molecular responses to one or more of the epithelial allergens, as indicated in Table 5.

Individuals with AD showed significantly higher serum titers (*p* < 0.05) to six epithelium molecules—Fel d 1, Fer d 7, Can f 1, Can f 2, Can f 4, and Can f 6—than subjects with AR and to four allergens—Fel d 1, Can f 1, Can f 2, and Can f 4—compared to the group of patients with the asthma phenotype. No quantitative significant differences were observed among AR, A, or AD in the mean levels of Fel d 2, Fel d 4, Can f 3, or Can f 5 in the studied population. Interestingly, 82 out of 84 subjects (97.61%) sensitized to epithelial molecules showed a concurrent sIgE response to 1 or more of the 17 assessed mite molecular allergens.

### 2.6. Pollen

A total of 64 subjects out of 150 (42.6%) showed a pattern of 31 different molecular responses to one or more of the following pollen allergens: Cyn d 1 (33.7%), Par j 2 (20.22%), Lol p 1 (17.97%), Art v 1 (13.48%), Cup a 1 (12.38%), Phl p 2 (10.11%), Pla a 3 (7.86%), Phl p 1 (5.61%), and Sal k 1, Ole e 1, and Bet v 6 (3.37%); Phl 5, Phl p 12, Art v 3, Bet v 2, and Pla a 1 (2.24%); and Phl p 6, Phl p 7, Pla a 2, and Bet v 1 (1.12%). No identification of Ole e 7 or Ole e 9 was spotted in the studied population (Table 6).

Twenty-eight subjects out of 64 were sensitized to a single molecular pollen allergen (43.75%). Cyn d 1 was confirmed as the most prevalent individual pollen allergen in 17 out of 64 patients (26.56%), followed by Par j 2, Art v 1, and Cup a 1 (4.68% in all 3 cases) and Phe p 1 (3.12%). Most individuals (56.25%) with a pollen sensitization displayed a concurrent response to two or more molecules. No quantitative significant differences were observed among AR, A, or AD in the mean levels of sIgE in 22 out 23 investigated pollen allergens. Only Cup a 1 showed significantly higher sIgE values (*p* = 0.0307) in patients with AD compared to those with AR.

Not only sensitization to pollen allergens was found in the present population. Sixty-three out of 64 subjects (98.43%) sensitized to any of the tested pollen molecules showed a coincident sIgE response to 1 or more of the 17 investigated mite allergens and the remaining subject—afflicted with AD—showed a complex pollen polysensitization pattern including Cup a 1, Pla a 2, Phl p 1, Lol p 1, Cyn d 1, and Art v 1 with epithelial allergens (Fel d 1).

### 2.7. Mold and Cockroach Allergens

Regarding molds, Alt a 1 and Alt a 6 were identified in only 11 (7.3%) and 1 (0.6%) out of 150 studied subjects, respectively. Not only sensitization to Alt a 1 or Alt a 6 was found in the current study population. All 12 (100%) subjects with a sensitization to Alt a 1 and/or Alt a 6 showed a concurrent sIgE response to 1 or more of the 17 mite allergens.

Cockroach allergens were also scarcely depicted, as Bla g 9 was only present in 10 individuals (6.6%), followed by Per a 7 in 8 individuals (5.3%) and Bla g 4 in 1 participant (0.6%). No sIgE response to Bla g 1, Bla g 2, or Bla g 5 was detected in the present population. All 18 (100%) sIgE responses to Bla g 9 and/or Per a 7 showed a concomitant polysensitization to 1 or more of the 17 included mite allergens.

## 3. Discussion

Climate change is already having an impact on the seasonality, production, and concentration, allergenicity, and geographic dissemination of airborne allergens, with subsequent consequences on human allergic disease [30,31]. Tenerife is the largest and highest oceanic island of the Canary archipelago, with a climate dominated by the influence of the cool humid northeast trade winds, associated with the Azores anticyclone [32]. Despite the local levels of air pollutants being rather low compared to continental regions of Europe and due to the proximity of the Canary Islands to the African continent, Saharan dust intrusions with PM_10_ concentrations higher than 50 μg/m^3^ have been independently related to affecting human health [33].

In the present study, the assessment of aeroallergen sensitization as a core biomarker for the classification of allergic disease [34], has consistently confirmed mites as the most prevalent allergen source affecting the local population. In fact, 96% of the investigated subjects confirmed a sensitization to ≥1 individual mite molecules, followed by a 56% sensitized to ≥1 epithelial allergen; 42.66% of patients had a sIgE response to ≥1 pollen molecules; and finally, 7.3% and 6.66% were sensitized to molds or cockroaches, respectively.

To date, more than 30 house dust mite (HDM) allergens have been described, with group 1 and 2 allergens leading the sIgE response in HDM-sensitized individuals globally [35]. Regarding the overall prevalence of individual allergens, in the present investigation, 11 molecules were considered serodominant (>50%) allergens in the studied population. Local subtropical conditions favoring an increased perennial exposure to both HDM and storage mites (SM) may explain the higher prevalence of mite sIgE responses in the studied population compared to results reported from Australia (77%) and Singapore (63%) [36,37].

Interestingly, despite these highly prevalent allergens inducing strong IgE responses, differences in the IgE antibody binding frequency have been addressed in both dissimilar and alike populations depending on the applied multiplex measurement platform [38,39]. In this regard, despite a former local specific allergen profile identifying identical prevalence rates for Der f 1, Der p 2, and Der f 2 using the ADVIA Centaur^®^ platform, Der f 2 and Der p 2 have repeatedly exhibited a higher sIgE binding frequency compared to group 1 HDM allergens with the multiplex ALEX^2^ array [40,41,42]. Such variations, according to the implemented multiplex assay in the overall prevalence of these representative HDM allergens, have also been addressed in former research from Spain, China, and Eastern Europe [43,44,45,46,47].

After HDM group 2 allergens, Der p 23 was identified as the third most frequent (82%) molecule in the studied population. Despite the high prevalence of sIgE binding to Der p 23, only one individual (0.66%) afflicted with AR showed a single sensitization to this molecule. In addition, our findings confirmed a prominent role for the so-called mid-tier HDM allergens—Der p 21, Der p 5, and Der p 7—as serodominant molecules in the studied population, especially in those subjects with A and AD. Moreover, in accordance with previously documented low cross-reactivity between group 5 allergens [48,49], molecular sensitization to Der p 5 (64.0%) was significantly (*p* < 0.05) increased compared to Blo t 5 (50.6%) across the study cohort, including in both respiratory and AD subjects. In addition, in line with our former observations, minor allergens such as tropomyosin and Der p 11 were scarcely represented in 10.6% and 2% of study patients, respectively, including patients affected by AD [50].

Although still under debate, it has been speculated that climate change in the last few decades may affect mite biology and that new allergens may become influential as sensitizing molecules [51]. Differently from *B. tropicalis*, which has been previously identified in nearly 24% of dust samples from local mattresses, *L. destructor*, *G. domesticus*, and *T. putrescetiae* were all detected in less than 6% of the studied samples before 2009 [52]. Interestingly, Lep de 2 was the fourth most frequently (76%) identified allergen in the current investigation, after Der f 2, Der p 2, and Der p 23, with more than 85% of these individuals concurrently sensitized to Gly d 2 and/or Tyr p 2, supporting a relevant IgE cross-reactivity among molecules [53] and a potential role as emerging allergen sources in this area.

Remarkably, epithelial allergens were confirmed to be the second most prevalent sensitization (56.0%) in our territory, after HDM and SM, with major allergens Fel d 1 (22.61%) and Can f 5 (15.47%) as leading individual molecules. Furthermore, as previous research has elegantly shown that structurally unrelated epithelial and mite allergens can activate epithelial cells through adjuvant-like protease-independent mechanisms [54], a coincident polysensitization pattern among epithelial and mite molecules was widely identified (97.61%), contributing to depicting the complexity of the indoor exposome affecting the current population. Moreover, perennial allergen sensitization has been related to an increased rate of asthma morbidity, including asthma exacerbations, hospital visits, and medication needs [12,55].

Pollen was shown to be the third most frequently identified molecule (42.6%), with an unexpected 33.7% of subjects showing a sensitization to Cyn d 1, followed by Par j 2 (20.22%), Lol p 1 (17.97%), Art v 1 (13.48%), and Cup a 1 (12.38%), which was not previously reported in our area. These data strongly support the inclusion of *Cynodon dactylon* in the local SPT battery of routine allergens. Concurrent sIgE responses to mite and/or epithelial allergens was found in all subjects (100%) with a pollen sensitization, giving insight into a coexisting local exposure to both perennial and seasonal molecules. Besides the exposure to pollen from the local vegetation, the island of Tenerife is also influenced by extra-regional pollen transport episodes from the Mediterranean region, the Saharan sector, and the Sahel, enclosing both tree and herb pollen [56]. Despite most pollen transport occurring within the island, pollen peaks may be markedly increased during specific meteorological situations driving air masses from areas where the plants providing the depicted pollen are present. In this regard, despite it having been estimated that up to 97% of annual counts of *Olea* pollen in Tenerife originates from extra-regional sources [57], Ole e 1 was identified in less than 4% of the present cohort.

Several clinical consequences may be outlined from this investigation. Firstly, the proposed panel accurately identified ≥98.0% of the molecular profile in patients from the same region but affected with distinct atopic diseases. Secondly, and in line with previous reports [58,59], we confirmed an increased number of sIgE binding molecules associated with the complexity of each underlying atopic phenotype. Moreover, significantly quantitatively higher titers (*p* < 0.05) were gradually found for major and mid-tier allergens in subjects with AD against those with A or AR, and in the group of asthmatics in respect to those with only AR. Transcutaneous sensitization to these allergens has previously been related to the pathogenesis of AD, which manifests with a genetically predetermined skin barrier defect, leading to an overexpression of pro-inflammatory cytokines and the subsequent activation of innate and adaptative immune responses [60]. In addition, as previously reported, a higher frequency of sIgE binding—especially to mite molecules—was found in the younger participants of the study, regardless of their underlying allergic condition [61,62,63].

A few limitations should be mentioned in the current study, as 4 out of 150 individuals (2.66%) with a positive SPT to local aeroallergens—3 asthmatics and 1 with AD—could not be identified by the multiplex array, and only a restricted number of predominantly white subjects from the same geographical region were finally included in the cross-sectional analysis. In addition, the study design did not allow for an investigation of how climate change impacts airborne allergens in terms of allergenicity, seasonality, production, and/or atmospheric concentration, as only transversal data were obtained from December 2021 to January 2023.

In line with former investigations, despite no specific pattern of component-sIgE sensitization being associated with any individual allergic disease, a broader repertoire of sIgE responses was related to an increased complexity of the clinical phenotype [36]. The identification of these highly prevalent molecules remains mandatory not only to describe part of the local external exposome but also to investigate their role as clinically relevant allergen sources and risk assessment in different allergic phenotypes [64]. In fact, the clinical relevance of only a few molecules present in the current assay—Der p 1, Der p 2, Der p 5, Blo t 5, Per a 10, Alt a 1, Fel d 1, Can f 1, Can f 2, Can f 3, Can f 4, and Can f 5—has been intrinsically proven, moving away from the traditional assumption that only allergens with high sIgE binding frequencies (>50%) are relevant [65].

## 4. Materials and Methods

### 4.1. Subjects

We consecutively recruited children and adult patients with an allergist-confirmed diagnosis of atopic disease—i.e., allergic rhinitis (AR), asthma (A), and/or atopic dermatitis (AD)—from the Outpatient Allergy Clinic and Severe Asthma Unit at Hospital Universitario de Canarias (Tenerife, Spain) from December 2021 to January 2023. The present investigation was previously evaluated and authorized by the domestic Ethical Committee and the corresponding informed consent documents were properly signed by all participants, as well as parents/guardians for those participants <18 y.o., upon inclusion in the study. Onset of referred clinical symptoms only after a minimum of 3 years of local residency was also required in order to meet the inclusion criteria in the present investigation. The severity and staging of allergic diseases were also clinically evaluated according to specific guidelines [26,27,28].

The following clinical data were collected from the patients’ medical records: sociodemographic data; clinical profile, including past medical conditions and current allergy diagnosis; and the characteristics of the associated medication, forced expiratory volume in the first second (FEV1), a validated Asthma Control Test (ACT), the SCORAD index, and SPT results. Following routine clinical practice, only subjects with a positive SPT and/or a specific sIgE to the corresponding aeroallergen extract—mites, pollens, molds, and/or animal fur epithelium—were included in the study. Patients under treatment with past or current allergen immunotherapy or monoclonal antibodies—biologics—were excluded. Pregnant and breastfeeding women were also excluded from the study.

### 4.2. Skin Prick Test

Percutaneous testing was carried out according to European standards [66], enclosing a diagnostic panel (Inmunotek, Madrid, Spain) with standardized raw extracts (*Dermatophagoides pteronyssinus* (*D. pteronyssinus*), *Blomia tropicalis* (*B. tropicalis*), *Lepidoglyphus destructor* (*L. destructor*), *Tyrophagus putrescentiae* (*T. putrescentiae*), cat and dog dander, grass mix (*Poa pratensis*, *Dactilis glomerata*, *Lolium perenne*, *Phleum pratense*, and *Festuca pratensis*), olive, *Parietaria judaica*, *Artemisa vulgaris*, *Alternaria alternata*, *Aspergillus fumigatus*, *Cladosporium herbarum*, and Blatella. Histamine (10 mg/mL) and saline were used as positive and negative controls as usual. Antihistamines were withdrawn a week before the SPT and wheal diameters were immediately measured after 20 min, with diameters greater than 3 mm regarded as positive.

### 4.3. Mite Allergenic Extracts

Proteins from mite bodies of *D. pteronyssinus*, *B. tropicalis*, and *L. destructor* were manufactured using standardized internal protocols (Inmunotek S.L., Madrid, Spain). The extracts were prepared by extracting the material in 0.01 M phosphate-buffered saline buffer (PBS; 1/5 wt:vol), pH 7.4, for 4–6 h at 5 ± 3 °C under magnetic stirring. Then, the extracts were centrifugated at 16,000× *g* for 30 min at 4 °C. Afterwards, the supernatants were recollected. The pellet was reconstituted and extracted in 0.01 M PBS (1/5 wt:vol) overnight at 5 ± 3 °C. Then, the extract was centrifugated again and the supernatant separated from the pellet. Both supernatants were mixed and subsequent dialysis was carried out by tangential ultrafiltration against highly purified water using Omega polyethersulfone membranes (TFF Cassette T series, Pall Life Sciences, Port Washington, NY, USA) with a pore size of 100 kDa. Finally, native extracts were frozen and lyophilized. The protein content was measured by the Bradford method.

### 4.4. SDS PAGE and IgE Western Blot

Proteins from *D. pteronyssinus B. tropicalis*, and *L. destructor* extracts were separated by 12% polyacrylamide gel with sodium dodecylsulfate (SDS-PAGE) under reducing conditions according to Laemmli’s method [67]. Proteins were visualized when the gel was stained with GelCode Blue stain reagent (Life Technologies, Carlsbad, CA, USA). For the Western blot, proteins from the gel electrophoresis were electrotransferred to 0.45 μm nitrocellulose membranes Bio-Rad, (Bio-Rad Laboratories, Hercules, CA, USA) and the binding of IgE antibody to allergens was analyzed using individual patients’ sera and anti-human IgE peroxidase conjugate (SouthernBiotech, Biotechnology Research, Birmingham, AL, USA). Chemiluminescence detection reagents (Western Lightning^®^ Plus-ECL, Perkin Elmer, Waltham, MA, USA) were added following the manufacturer’s instructions and the image was analyzed in Image Lab Touch software 3.0.1.14. IgE binding bands were identified using the BioRad Diversity database program (Bio-Rad Laboratories, Hercules, CA, USA).

### 4.5. Serological Analysis

Blood samples were obtained from all participating individuals, identified with a code label, stored at −40 °C, and thawed immediately prior to the in vitro assay. Total IgE levels and sIgE were measured (ALEX MacroArray Diagnostics, Vienna, Austria) according to the manufacturer’s instructions in all included subjects. In brief, ALEX is a multiplex array containing 282 reagents (157 whole allergens and 125 molecular components). The different allergens and components are coupled onto polystyrene nano-beads, and then the allergen beads are deposited onto a nitrocellulose membrane, as formerly published [68]. Total IgE levels were expressed in international units per unit volume (IU/mL), and sIgE levels were expressed in kU_A_/L. Values ≥ 0.35 kU_A_/L were considered positive. A total of 17 mite molecular allergens were included: Der p 1, Der p 2, Der p 5, Der p 7, Der p 10, Der p 11, Der p 20, Der p 21, Der p23, Der f 1, Der f 2, Blo t 5, Blot 10, Blo t 21, Lep d 2, Gly d 2, and Tyr p 2. A total of 10 cat and dog epithelial molecular allergens were included: Fel d 1, Fel d 2, Fel d 4, Fel d 7, Can f 1, Can f 2, Can f 3, Can f 4, Can f 5, and Can f 6. A total of 23 pollen allergens were included: Bet v 1, Bet v 2, Bet v 6, Cup a 1, Pla a 1, Pla a 2, Pla a 3, Ole e 1, Ole e 7, Ole e 9, Phl p 1, Phl p 2, Phl p 5, Phl p 6, Phl p 7, Phl p 12, Lol p 1, Cyn d 1, Sal k 1, Pla l 1, Par j 2, Art v 1, and Art v 3. Alt a 1 and Alt a 6 were investigated for mold sensitization, whereas Bla g 1, Bla g 2, Bla g 4, Bla g 5, Bla g 9, and Per a 7 were designed to evaluate cockroach sensitization.

### 4.6. Statistical Analysis

Demographic features were summarized by medians and standard deviations for continuous variables and percentages for categorical variables. To compare differences, analysis of variance, Kruskal–Wallis, Mann–Whitney U, and Chi-square tests are required for parametric continuous, nonparametric continuous, and categorical variables, respectively. A *p*-value of less than 0.05 was considered statistically significant. All statistical data were analyzed using GraphPad Prism version 8.0.0 for Windows, GraphPad Software, La Jolla, CA, USA.

## 5. Conclusions

The present study is, to the best of our knowledge, the first to investigate a real-life global molecular response to a comprehensive panel of aeroallergens in our local territory in subjects afflicted with different atopic conditions. The outlined PAMD@ panel acted as a sensitive and highly specific tool for a precise molecular diagnosis in allergic patients strongly influenced by a complex external exposome.

Despite our results confirming an increasingly pleomorphic profile of sIgE binding molecules in connection with the complexity of the allergic phenotype, no single molecular allergen could be identified as a surrogate marker for any of the studied atopic conditions. In relation to the aggregation of molecules and evolving profiles, concurrent polysensitization to unrelated indoor allergenic sources—especially mites and epithelia—were broadly predominant, making it challenging to propose an individualized treatment—i.e., allergen immunotherapy—for most patients.

Finally, as climate change impacts airborne allergens in terms of allergenicity, seasonality, production, and atmospheric concentration—leading to subsequent outcomes in human health—the identification of the local external exposome stands out for a personalized diagnosis and therapy in clinically distinctive allergic phenotypes.

## Figures and Tables

**Figure 1 ijms-24-10467-f001:**
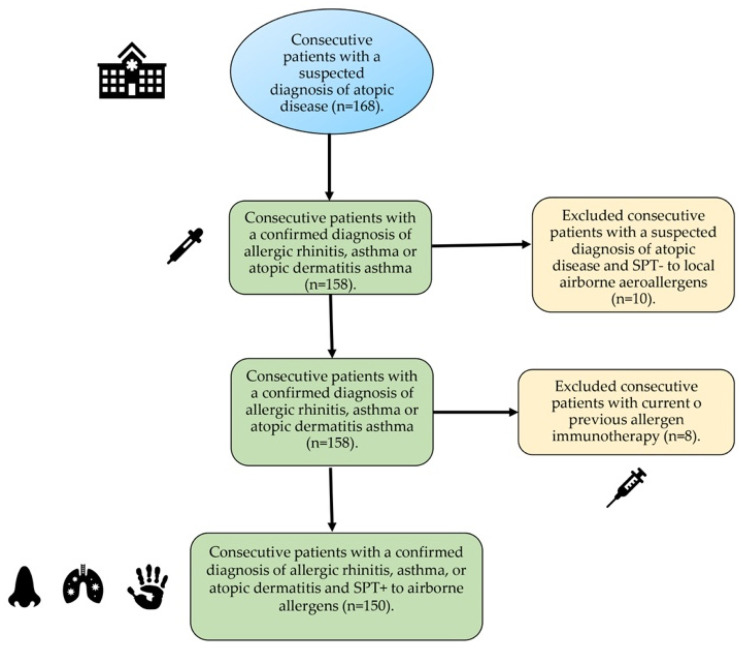
Flow diagram of study selection.

**Figure 2 ijms-24-10467-f002:**
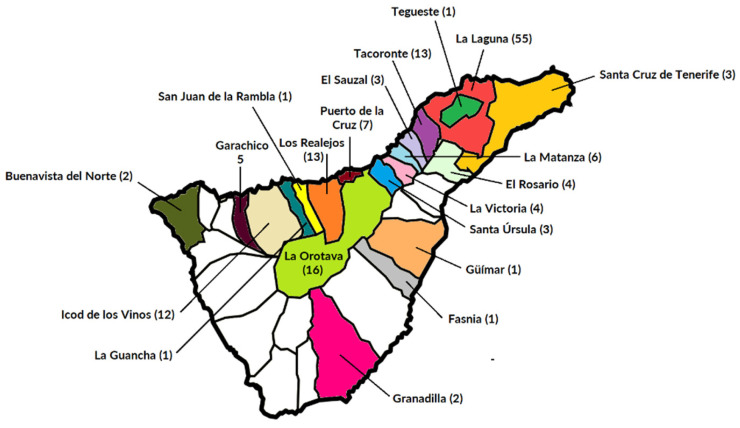
Distribution of patients according to their local municipality on the island of Tenerife, Spain (28°16′7″ N, 16°36′20″ W). Onset of referred clinical symptoms after a minimum of 3 years of local residency was required to meet the inclusion criteria in the present investigation.

**Figure 3 ijms-24-10467-f003:**
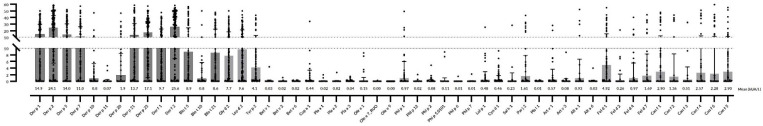
Specific IgE serodominance in a comprehensive panel of molecular aeroallergens.

**Figure 4 ijms-24-10467-f004:**
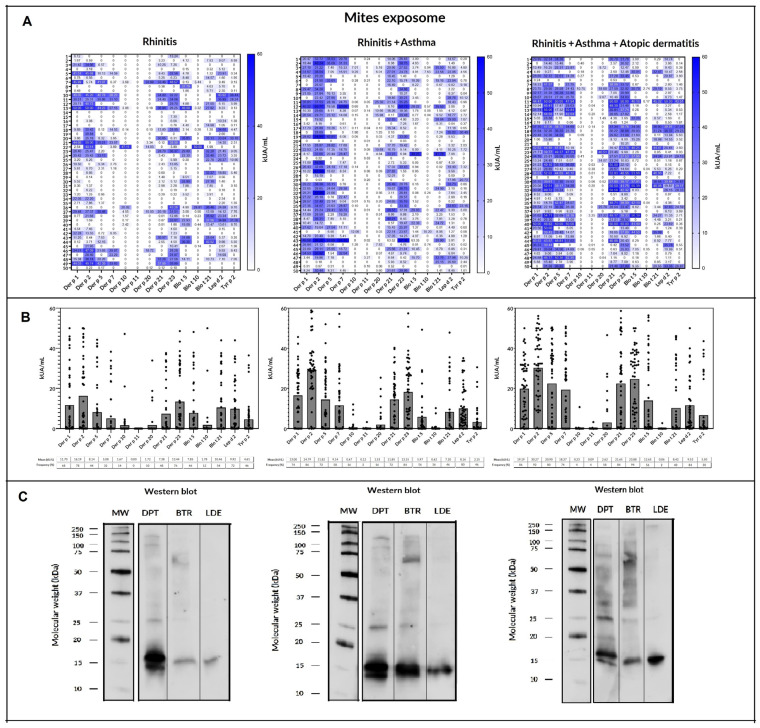
Sensitization profile to specific IgE (sIgE): (**A**) heatmap and (**B**) scatter plots with bars of a comprehensive panel of molecular allergens of *Dermatophagoides pteronyssinus*, *Blomia tropicalis, Lepidoglyphus destructor*, and *Tyrophagus putrescentiae* in 50 rhinitis, 50 rhinitis and asthma, and 50 rhinitis, asthma, and atopic dermatitis patients. (**C**) IgE: Western blot of the different groups included against *Dermatophagoides pteronyssinus* (DPT), *Blomia tropicalis* (BTR), and *Lepidoglyphus destructor* (LDE). Broadly different patterns of sIgE binding were identified for each group of patients.

**Table 1 ijms-24-10467-t001:** Descriptive statistics regarding basal comorbid conditions and associated clinical features of the studied population (*n* = 150).

	Allergic Rhinitis	Allergic Asthma	Atopic Dermatitis
***n* = 150**	50	50	50
**Age (y.o.) median (range)**	27.5 (9–70)	32 (8–67)	26 (9–62)
**<20 y.o. (*n* = 54)**	22 (40.7%)	16 (29.6%)	16 (29.6%)
**≥20 y.o. (*n* = 96)**	28 (29.1%)	34 (35.4%)	34 (35.4%)
**Sex (F/M)**	29/21	32/18	22/28
**Mild–moderate atopic disease**	23	29	28
**Severe atopic disease**	27	21	22
**Food allergy (*n* = 23)**	8 (34.7%)	6 (20.0%)	9 (39.1%)
**Drug allergy (*n* = 11)**	2 (18.1%)	6 (54.5%)	3 (27.27%)
**SPT+ to any aeroallergen**	50 (100%)	50 (100%)	50 (100%)
**Total IgE (IU/mL) median (range)**	56 (23.88–1083)	674 (56.5–19,993)	898 (81.81–17,420)
**Blood eosinophils/mm^3^ median (range)**	340 (140–530)	365 (20–1290)	375 (25–1880)
**Family history of atopy**	36 (72%)	38 (76%)	39 (78%)

SPT: skin prick test. Median values (ranges) are shown.

**Table 2 ijms-24-10467-t002:** Prevalence of sensitization to grouped local aeroallergens by the skin prick test (*n* = 150).

Positive SPT	*N* (%)
HDM and/or SM	146 (97.33)
Cat and/or dog dander	46 (30.66)
Pollen	14 (9.33)
Cockroach	9 (6)
Molds	8 (5.33)

SPT: skin prick test. HDM: house dust mites. SM: storage mites.

**Table 3 ijms-24-10467-t003:** Serological analysis of specific IgE (sIgE) responses (kU/L) to mite molecular allergens in patients with allergic rhinitis (AR, *n* = 50), asthma (A, *n* = 50), or atopic dermatitis (AD, *n* = 50). The number (%) of subjects (*n* = 150) sensitized to the corresponding mite molecular allergen is shown. Bold figures indicate quantitative significant differences (*p* < 0.05) in mean sIgE to mite molecular allergens among atopic conditions.

Mite Allergen	Mean sIgE (M ± SD)	No. of Sensitized Patients (All)	Mean sIgE in AR (M ± SD)	No. of Sensitized Patients (AR)	Mean sIgE in Asthma (M ± SD)	No. of Sensitized Patients (A)	Mean sIgE in Atopic Dermatitis (M ± SD)	No. of Sensitized Patients (AD)
**Der f 2**	25.6 ± 14.2	126 (84)	**17.17 ± 0.2**	39 (78)	**26.2 ± 11.41**	43 (86)	**32.7 ± 0.31**	46 (92)
**Der p 2**	24.1 ± 15.3	125 (83)	**16.19 ± 0.1**	39 (78)	**24.79 ± 12.3**	43 (86)	**30.27 ± 2.96**	46 (92)
**Der p 23**	17.15 ± 0.3	123 (82)	**13.44 ± 1.2**	38 (76)	**13.15 ± 6.25**	42 (84)	**23.88 ± 12.7**	47 (94)
**Lep d 2**	9.68 ± 1.5	114 (76)	9.92 ± 0.2	36 (72)	8.16 ± 5.23	40 (80)	9.53 ± 7.47	42 (84)
**Der p 1**	14.9 ± 3.34	109 (72.6)	**11.7 ± 4.3**	34 (68)	13 ± 9.3	37 (74)	**19.19 ± 14.5**	43 (86)
**Der f 1**	9.79 ± 2.51	104 (69.3)	**3.63 ± 0.89**	26 (52)	**10.36 ± 8.32**	38 (76)	**15.2 ± 4.1**	43 (86)
**Gly d 2**	7.72 ± 1.53	101 (67.3)	7.34 ± 0.34	37 (74)	5.03 ± 3.9	32 (64)	9.5 ± 3.81	36 (72)
**Der p 21**	13.71 ± 3.6	100 (66.6)	**7.38 ± 0.36**	24 (48)	**11.85 ± 4.17**	36 (72)	**21.65 ± 0.22**	42 (84)
**Der p 5**	14.07 ± 1.0	96 (64)	**8.14 ± 0.72**	22 (44)	**11.82 ± 1.66**	36 (72)	**20.9 ± 2.3**	40 (80)
**Der p 7**	11.01 ± 1.9	82 (54.6)	**5.08 ± 0.39**	16 (32)	**9.14 ± 4.21**	29 (58)	**18.37 ± 1.9**	37 (74)
**Blo t 5**	8.96 ± 2.84	76 (50.6)	7.85 ± 0.53	23 (46)	5.97 ± 0.2	28 (56)	12.65 ± 2.75	28 (56)
**Tyr p 2**	4.18 ± 0.2	68 (45.3)	4.61 ± 1,02	23 (46)	2.15 ± 2	23 (46)	5.5 ± 0.22	25 (50)
**Blo t 21**	8.64 ± 0.31	66 (44)	10.46 ± 0.8	27 (54)	7.1 ± 5.3	23 (46)	8.42 ± 0.1	20 (40)
**Der p 20**	1.94 ± 0.5	22 (14.6)	1.72 ± 0.08	5 (10)	1.53 ± 0.09	8 (16)	2.62 ± 0.5	9 (18)
**Der p 10**	0.86 ± 0.23	16 (10.6)	1.67 ± 0.06	7 (14)	0.67 ± 2.3	8 (16)	<0.35	2 (4)
**Blo t 10**	0.83 ± 0.41	14 (9.3)	1.78 ± 0.09	6 (12)	0.62 ± 0.2	8 (16)	<0.35	1 (2)
**Der p 11**	0.75 ± 0.3	3 (2.0)	<0.35	0 (0)	<0.35	1 (2)	<0.35	2 (4)

**Table 4 ijms-24-10467-t004:** Number of identified mite molecular allergens and corresponding basal atopic disease (allergic rhinitis, asthma, and atopic dermatitis) in 150 patients studied via microarray. Most subjects (64%) with allergic rhinitis displayed a specific IgE response < 9 mite molecules, in contrast to patients with asthma (58%) or atopic dermatitis (68%), who showed a polysensitization profile to ≥10 individual mite allergens.

Number of Identified Mite Allergens	Allergic Rhinitis	Asthma	Atopic Dermatitis
0	0	3	1
1	1	1	2
2	0	1	0
3	2	1	1
4	5	1	0
5	3	0	0
6	5	4	1
7	6	1	0
8	3	4	5
9	7	5	5
10	4	3	5
11	6	11	11
12	5	5	10
13	1	7	7
14	1	2	0
15	1	0	1
16	0	1	1
17	0	0	0

**Table 5 ijms-24-10467-t005:** Specific IgE profiles aggregated into selected cat and dog epithelial allergens in 84 out of 150 subjects tested via microarray. Profiles are ordered by the increasing number of recognized molecules. Asterisks (*) indicate specific IgE sensitization to a single cat and/or dog molecular allergen.

*n* = 84	%	Number of Molecules	Fel d 1	Fel d 2	Fel d 4	Fel d 7	Can f 1	Can f 2	Can f 3	Can f 4	Can f 5	Can f 6
19	22.6	1	*									
1	1.1	1				*						
13	15.4	1									*	
2	2.3	2	*		*							
11	13.0	2	*								*	
3	3.5	2								*	*	
4	4.7	2	*							*		
1	1.1	3	*				*			*		
1	1.1	3	*		*							*
1	1.1	3	*			*	*					
2	2.3	3	*							*	*	
1	1.1	3	*	*							*	
2	2.3	3				*	*				*	
1	1.1	3	*						*		*	
1	1.1	3				*	*			*		
3	3.5	4	*			*	*				*	
1	1.1	4	*		*		*				*	
1	1.1	4	*		*	*	*					
2	2.3	4	*				*			*	*	
1	1.1	4	*		*	*					*	
1	1.1	4	*			*	*			*		
1	1.1	5	*	*	*		*		*			
1	1.1	5	*		*	*	*				*	
1	1.1	6	*		*	*	*				*	*
1	1.1	6	*	*	*	*	*		*			
1	1.1	7			*	*	*	*		*	*	*
1	1.1	7	*			*	*	*		*	*	*
2	2.3	8	*		*	*	*	*		*	*	*
1	1.5	8	*	*		*	*	*	*	*		*
1	1.1	8	*			*	*	*	*	*	*	*
2	2.3	9	*	*	*	*	*	*	*	*		*

**Table 6 ijms-24-10467-t006:** Specific IgE profiles aggregated into selected pollen allergens in 64 out of 150 subjects tested via microarray. Profiles are ordered by increasing number of recognized molecules. Asterisks (*) indicate specific IgE sensitization to a single pollen molecular allergen.

*n* = 64	%	Number of Molecules	Cyn d 1	Par j 2	Phl p 1	Phl p 2	Phl p 12	Lol p 1	Art v 1	Art v 3	Cup a 1	Sal k 1
3	4.6	1						*				
3	4.6	1							*			
2	3.1	1			*							
3	4.6	1		*								
17	26.5	1	*									
2	3.1	1					*					
3	4.6	1									*	
3	4.6	2			*			*				
1	1.5	2		*							*	
1	1.5	2	*						*			
1	1.5	2						*			*	
1	1.5	2		*				*				
2	3.1	2		*					*			
1	1.5	2		*		*						
2	3.1	2									*	*
1	1.5	2	*							*		
1	1.5	2	*						*			
1	1.5	2	*	*								
1	1.5	2		*						*		
1	1.5	3	*		*			*				
2	3.1	3	*	*					*			
1	1.5	3			*			*			*	
1	1.5	3	*			*						*
1	1.5	3	*					*	*			
1	1.5	3				*					*	*
3	4.6	4	*		*	*		*				
1	1.5	4			*			*			*	*
1	1.5	4		*	*	*					*	
1	1.5	4			*	*			*		*	
1	1.5	4	*	*	*				*			
1	1.5	5		*	*	*		*	*			

## Data Availability

The data that support the findings of this study are available from Servicio Canario de Salud, but restrictions apply to the availability of these data, which were used under license for the current study and thus are not publicly available. Data are, however, available from the authors upon reasonable request and with the permission of the Servicio Canario de Salud.

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
