# Peer review of "Molecular Mapping of Allergen Exposome among Different Atopic Phenotypes"

_ijms, 2023, doi:10.3390/ijms241310467_

Round 1

Reviewer 1 Report

Personalized diagnosis and therapy is the new way that captures the current trends and so this article brings relevant information in the field.

Author Response

The Authors would like to thank Reviewer´s  1 valuable comments. 

Reviewer 2 Report

Authors wished to investigate -through a personalized Precision Allergy Molecular Diagnosis model approach- the molecular sensitization map to aeroallergens  in three allergic conditions: allergic rhinitis (n=50), allergic asthma (n=50) and atopic dermatitis (n=50) in a subtropical climate conditions (Tenerife, Spain).

Some points need to be addressed in a revised version of the manuscript.

Title and Introduction. The interaction between climate change, exposome and sensitization has to be studied by comparing countries with different climatic conditions or historical observations in the same country. A recent study, which the Authors should cite, properly explores this aspect.( A molecular sensitization map of European children reveals exposome- and climate-dependent sensitization profiles. Allergy 2023).

Methods. It is unclear how an equal number of patients (n=50) with the three allergic diseases can be obtained by screening 168 patients with suspected allergic disease

Table 1. The range of total IgE observed in patients with asthma includes abnormally elevated values, highly unusual in asthma, unless complicated by allergic broncho-pulmonary aspergillosis

Table 2 and Table 5..

SPTs have been found positive in 14/150 subjects, while specific IgE for pollen molecules have been found in 64/150 subjects, as reported in Table 5. Please comment on this conflicting observation.

Discussion. The study design is not adequate to investigate how climate change impacts on airborne allergens in terms of allergenicity, seasonality, production, and atmospheric concentration. Authors should  expand the discussion on the different molecular sensitization patterns in the three allergic diseases, differences that do not seem striking.

Minor editing of English language required

Author Response

The Authors would like to thank Reviewer´s 2 for the accurate and valuable comments. 

Reviewer 3 Report

The authors analyzed the allergic exposome in different atopic conditions. The study is of interest, however, I have several concerns, that need to be addressed by the authors.

1. Please provide more data about the patient population: ethnic origin, place of living (countryside vs. cities vs. towns)

2. Based on the authors' statement, AD was diagnosed according to SCORAD. In fact, SCORAD is the assessment of AD severity, not a diagnostic tool. Please provide details, on how the AD diagnosis was established. 

3. No information is provided about the severity of the diseases among studied subject.

4. Little information is provided about the current treatment in AR and AD patients

Some minor spelling mistakes should be corrected by the authors. 

Author Response

The Authors would like to thank Reviewers´ 3 valuable suggestions.

Round 2

Reviewer 2 Report

The revised version of the manuscript addressed the critical points of the first version of the manuscript.

none

Reviewer 3 Report

Thank you for correcting your manuscript.